# The Influence of a Hyperglycemic Condition on the Population of Somatostatin Enteric Neurons in the Porcine Gastrointestinal Tract

**DOI:** 10.3390/ani10010142

**Published:** 2020-01-15

**Authors:** Michał Bulc, Katarzyna Palus, Jarosław Całka

**Affiliations:** Department of Clinical Physiology, Faculty of Veterinary Medicine, University of Warmia and Mazury, Oczapowskiego Str. 13, 10-719 Olsztyn, Poland; katarzyna.palus@uwm.edu.pl (K.P.); jaroslaw.calka@uwm.edu.pl (J.C.)

**Keywords:** somatostatin, hyperglycemia, enteric neurons, pig

## Abstract

**Simple Summary:**

Diabetes mellitus is a chronic metabolic disorder, in the course of which prolonged episodes of hyperglycemia are common. Hyperglycemia leads to damage and disruption in the proper function of many organs, including the gastrointestinal tract. Alimentary tract functions are regulated by triple innervation, in which the enteric nervous system plays a dominant role. Appropriate digestive functions are controlled by numerous biologically active substances which are synthesized and released through enteric neurons. One of the most common substances that occurs in the gastrointestinal tract is somatostatin. This peptide is involved in the regulation of a wide range of digestive functions, such as intestinal motility, secretory activity, and others. Furthermore, somatostatin participates in sensory and pain stimuli as well as modulation and release of other active factors. Somatostatin is also involved in the response of neurons to many pathological conditions. In this study, it was found that somatostatin secreted by enteric neurons has an important effect in hyperglycemic conditions induced by streptozotocin injection in the swine diabetes model. These data could provide useful information for further investigations into the functions of neuroactive substances in digestive tract pathophysiology in the course of diabetes.

**Abstract:**

Somatostatin (SOM) is the most common agent in the gastrointestinal (GI) tract that is involved in the regulation of several gastric functions, as well as in gastric disorders. Hyperglycemia, which develops as a consequence of improperly treated diabetes, can cause numerous disturbances in the appropriate functioning of the gastrointestinal tract. High glucose level is toxic to neurons. One of the lines of defense of neurons against this glucotoxicity are changes in their chemical coding. To better understood the role of SOM secreted by enteric neurons in neuronal response on elevated glucose level, pancreatic β cells were destroyed using streptozotocin. Due to the close similarity of the pig to humans, especially the GI tract, the current study used pigs as an animal model. The results revealed that the number of enteric neurons immunoreactive to SOM (SOM-IR) in a physiological state clearly depend on the part of the GI tract studied. In turn, experimentally induced diabetes caused changes in the number of SOM-IR neurons. The least visible changes were observed in the stomach, where an increase in SOM-IR neurons was observed, only in the submucosal plexus in the corpus. However, diabetes led to an increase in the population of myenteric and submucosal neurons immunoreactive to SOM in all segments of the small intestine. The opposite situation occurred in the descending colon, where a decrease in the number of SOM-IR neurons was visible. This study underlines the significant role of SOM expressed in enteric nervous system neurons during diabetes.

## 1. Introduction

The alimentary tract is capable of autonomic control of motor activity, fluid secretion, as well as efficient food digestion and absorption [1]. Functional independence of the gastrointestinal (GI) tract is possible due to the presence of unique innervation [1,2]. In particular, structures of the GI tract are provided with both extrinsic and intrinsic innervation [3]. Extrinsic innervation includes the vagal parasympathetic components. This part of innervation is able to control afferent sensitive and pain stimuli information as well as efferent secretomotor activity. Moreover, sympathetic innervation also regulates GI tract functions [3], although a characteristic feature of the alimentary tract is the presence of intrinsic innervation represented by the enteric nervous system (ENS) [4]. The ENS, also called “the second brain”, creates the dense mesh network of neurons and glial cells (starting in the esophagus and ending in the rectum) which controls most of the GI tract functions [5]. The ENS is assembled into the GI tract wall and is organized in several plexuses. Depending on the GI tract region as well as animal species, enteric neurons are organized in two or three intramural ganglia [6]. For example, in rodents, two kinds of enteric ganglions are distinguished. The myenteric ganglion is located between longitudinal and circular muscle layers. Moreover, each ganglion is connected to another by a dense network of nerve fibers, thus, creating the myenteric plexus (MP). It is worth underlining that MP is present in all mammalian species. Furthermore, it occurs in all segments of the digestive tract [7]. The second kind of ganglion, called the submucosal ganglion, is located on the border submucosal layer near the lamina propria. All submucosal ganglia connected by nerve fibrer create the submucosal plexus (SP). In rodents, SP is not present in the stomach and occurs only in the small and large bowels. To the contrary, in large animals (including humans), SP is present in all GI tract segments. Additionally, in the intestine, it is divided into an outer submucous plexus (OSP) which is located near the internal side of the circular muscle layer, and the inner submucous plexus (ISP) which is positioned near the lamina propria of the mucosal layer [6,7].

Enteric neurons, comprised of more than 15 functional classes, utilize a wide range of neurotransmitters [8]. Due to the type of neurotransmitter secreted, ENS neurons are divided into three classes, excitatory, inhibitory, and sensory neurons. This classification is, of course, very theoretical because an individual enteric neuron can synthesize more than one active substance [4,9]. One of the commonly found substances in the GI tract is somatostatin (SOM). SOM is classified as an inhibitory hormone. Its action is limited to different parts of the GI tract as well as the central and peripheral nervous systems [10,11]. In the GI tract, somatostatin suppresses the release of gastrointestinal hormones, decreases the rate of gastric emptying, and reduces smooth muscle contractions and blood flow within the intestine. Moreover, suppression of the exocrine secretory activity of the pancreas and inhibition of the release of glucagon and insulin from pancreatic islets are also described as SOM action [12]. Its presence has been described in different mammalian species, although interspecies differences in the exact distribution of this peptide in the ENS have been clearly visible [10,13,14]. Nevertheless, the presence of SOM has been observed both in the enteric plexuses, as well as in all segments of the alimentary tract [10].

One of the specific features of neurons, in both the central and peripheral nervous systems, is the ability to adapt to changes that occur in the body. This property is defined as a neuroplasticity. This is a multidirectional concept that consists of changes in the morphology of neurons, such as modification of shape and the length of processes. Moreover, the variability of electrophysiological properties was also observed [10]. A very important issue in neuroplasticity is variation, including changes in the neurochemical component of enteric neurons [15,16]. The list of diseases that affect the chemical changes of neurons, including enteric neurons, is very long. One of these afflictions is diabetes, a metabolic disease which damages many organs, including the GI tract [17,18]. Diabetic complications from the nervous system contribute to a significant reduction in the quality of life and are a cause of increased patient mortality [19]. All clinical symptoms of diabetic complications are well known, but the role of neuropeptides synthesized in neurons, especially in enteric neurons, in diabetic complications is not fully understood [19]. SOM is a biologically active enteric substance that undergoes changes during numerous gastric disturbances. Previous studies have documented the influence of axotomy, experimentally induced inflammation, and *Bacteroides fragilis* infection on the neurochemical properties of SOM in enteric neurons [10,20,21]. There are few reports in the literature describing the effect of streptozotocin-induced diabetes on somatostatin containing enteric neurons [22,23], and it should be noted that such research was conducted only on selected fragments of the GI tract using rodents as an animal model [22,24]. Therefore, in this study, the effect of streptozotocin-induced diabetes on the distribution and number of intramural neurons (in the stomach, small intestine, and descending colon) containing SOM was examined. Swine were used as the animal model due to anatomical, histological, and physiological similarities to humans, which is considered to be a better model than rodents [22,25,26,27]. The obtained data of the influence of diabetes on SOM immunoreactivity in the ENS sheds light on the involvement of this neuropeptide on the development of diabetic gastrointestinal complications in humans. Therefore, further clinical studies could help to improve the treatment of diabetes complications and to improve the quality of life in people suffering from diabetes.

## 2. Materials and Methods

All experiments were approved by the Local Ethical Committee in Olsztyn (Poland) (decision number 13/2015/DTN) and according to the Act for the Protection of Animals for Scientific or Educational Purposes of 15 January 2015 (Official Gazette 2015, no. 266), applicable in the Republic of Poland with special attention paid to minimizing any stress reaction. These studies were performed on ten juvenile female pigs of the White Large Polish breed, weighing 17 to 20 kg and 12 weeks old at the beginning of the experiment. The animals were obtained from a commercial pig farm. After an acclimatization period (one week), the pigs were randomly divided into two groups, control (N = 5) and experimental (N = 5). In animals from the experimental group, diabetes was induced as previously described [28,29]. For induction of diabetes, streptozotocin (STZ) (Sigma-Aldrich, St. Louis, MO, USA, S0130) 150 mg/kg of body weight, dissolved in a freshly prepared disodium citrate buffer solution (pH 4.2), 1 g streptozotocin/10 mL solution) was used. Before STZ injection, the pigs were anesthetized and an intravenous needle was then inserted into an ear blood vessel and streptozotocin was injected with an ongoing infusion for roughly 5 min. Since side effects often occur after diabetes induction, animals were fasted for 18 h before the experiment. The control pigs were injected with equal amounts of vehicle (citrate buffer). Due to the possibility of severe hypoglycemia due to pancreatic islet necrosis and the release of large amounts of insulin, 250 mL of 50% glucose solution per animal was given. After inducing diabetes, the animals in both groups were kept under standard laboratory conditions, were fed standard fodder (rapeseed meal 6.0%, soybean meal 9.0%, wheat 54.0%, barley 28.5%, and others 2.5%) and had free access to tap water. The blood glucose level was evaluated using an Accent-200 (Berlin, Germany) biochemical analyzer, with a colorimetric measurement at a wavelength of 510 nm or 670 nm. The plasma glucose level was measured from blood collected from the ear. Measurements were made at the beginning of the experiment and subsequent 48 h after the injection of streptozotocin. Measurements of glucose levels were then performed weekly until the end of the experiment.

Six weeks after diabetes induction, animals with both groups were anesthetized via intravenous administration of pentobarbital (Vetbutal, Biowet, Poland) and perfused transcardially via the ascending aorta with freshly prepared 4% paraformaldehyde in 0.1 M (molar) phosphate buffer (PB pH 7.4). All segments of the alimentary tract were removed. From the stomach (antrum, corpus, and pylorus), duodenum, jejunum (approximately 20 cm from the duodenum), ileum (approximately 4 cm before the caecum) and descending colon, approximately 2 cm were collected, post-fixed by immersion in the same fixative for 10 min, then washed with 0.1 M PB (pH 7.4) over two days and finally transferred and stored at 4 °C in an 18% buffered sucrose solution (pH 7.4) containing 0.001% natrium azide. The tissue blocks were cut in frontal or sagittal planes using a Microm HM 560 cryostat (Carl Zeiss, Berlin, Germany) at a thickness of 12 μm and mounted on gelatinized glass slides.

The sections were processed for a routine double immunofluorescence technique. After drying at 32 °C for 45 min, the sections were rinsed in phosphate buffer containing 0.8% of sodium chloride and 0.02% of potassium chloride (PBS, 3 × 10 min) and incubated in 10% horse serum in PB with 0.3% Triton X-100 and 1% bovine serum albumin (BSA) for 20 min. The sections were then incubated overnight at room temperature in a humid chamber with primary antibodies (SOM, rat monoclonal, Biogenesis, Suffolk, UK, 1:100) diluted in PBS containing 0.3% Triton X-100 and 1% BSA and Hu C/D (mouse polyclonal, Invitrogen, Waltham, MA, USA, cat # A-212711:1.000, working dilution 1:1000). On the following day, the sections were rinsed (PBS, 5 × 15 min) and incubated with secondary antibodies (donkey anti-mouse Alexa Fluor 488, 1:1000 Invitrogen, Waltham, MA, USA and donkey anti-rat Alexa Fluor 546 1:1000 Invitrogen, Waltham, MA, USA) diluted in PBS containing 0.25% BSA and 0.1% Triton X-100) for 4 h. The sections were then rinsed (PBS, 3 × 5 min) and covered with a polyethylene glycol/glycerine solution.

Double-labeled perikarya were evaluated under an Olympus BX51 microscope equipped with epifluorescence and appropriate filter sets, coupled with a digital monochromatic camera (Olympus XM 10, Tokyo, Japan) connected to a PC and analyzed with cellSens Dimension Image Processing software (Olympus, Tokyo, Japan). For determination of the percentage of SOM-IR neurons, at least 500 perikarya with clearly visible nucleus immunoreactive to Hu C/D in the particular type of enteric plexuses from each animal were investigated for the occurrence of SOM. The obtained results were pooled and presented as the mean standard error (±SEM). To avoid double-counting the same perikarya, the investigated sections of intestine were located at least 100 µm apart. The data pooled from all animal groups were statistically analyzed using Statistica 13 software (StatSoft Inc., Tulsa, OK, USA) and expressed as a mean ± SEM. Significant differences were evaluated using Student’s *t*-test for independent samples (* *p* < 0.05, ** *p* < 0.01, and *** *p* < 0.001).

## 3. Results

### 3.1. General Animal Conditions and Glucose Serum Level

Two days after streptozotocin injection, the pigs in the experimental group had significant hyperglycemia. Nevertheless, all animals with diabetes survived the experiment in good general condition. Moreover, despite the high glucose level, no animals required exogenous insulin injection. At the beginning of the experiment, immediately before streptozotocin injection, the mean glycaemia level in all animals was within standard reference values for the pig (5.01 mmol/L ± 0.10 mmol/L) (Figure 1). After streptozotocin injection, systematic growth of the mean glucose level was observed. Generally, the baseline glucose level in diabetic pigs was at a level exceeding 20 mmol/L and remained unchanged at the end of the experiment. This information was also described in previous studies [28,29]. All results of the glucose level are presented in Figure 1.

### 3.2. Stomach

#### 3.2.1. Myenteric Plexus (MP)

During this study, SOM-IR perikarya were present in the myenteric plexuses of the corpus and pylorus, while in the antrum they were not observed (Figure 2A and Figure 3A). Generally, the population of SOM-IR enteric neurons in the stomach was not very numerous (1.33 ± 0.12% in the corpus and 1.20 ± 0.14% in the pylorus) (Figure 2A and Figure 3B,C). In addition, the diabetic condition did not change the number of enteric neurons immunopositive to SOM in all investigated areas of the stomach (0.21 ± 0.07% in the antrum; 1.56 ± 0.51% in the corpus and 1.15 ± 0.24% in the pylorus) (Figure 2A and Figure 3D–F). 

#### 3.2.2. Submucosal Plexus

Since the submucosal plexus present in the stomach is not divided into outer and inner parts, it was described as a submucosal plexus (Figure 2B). In the antrum and pylorus, the number of neurons within the submucosal plexus was insufficient for statistical analysis. In the corpus, the number of neurons expressing SOM was estimated at 2.15 ± 0.26%. In the experimental group, a high glucose level produced changes in the distribution of SOM-IR neurons within the submucosal plexus in the corpus, i.e., an increased number of SOM-IR neurons (to 6.78 ± 1.17%) were observed.

### 3.3. Small Intestine

#### 3.3.1. Myenteric Plexus (MP)

SOM-IR structures were present in all investigated parts of the porcine small intestine under physiological conditions as well as in the experimental group. However, the number of SOM-expressing neurons differed between particular fragments of the small bowel (Figure 4A and Figure 5). The smallest number of SOM positive neurons were present in the duodenum (0.35 ± 0.18%), slightly more neurons were observed in the jejunum (1.29 ± 0.22%), while in the ileum the number of SOM-IR neurons was estimated at 2.69 ± 0.78%. Chemical-induced diabetes caused a considerable increase in SOM immunoreactivity only in the ileum (to 5.90 ± 0.98%) (Figure 4A–C and Figure 5A–F).

#### 3.3.2. Inner and Outer Submucosal Plexuses (ISP and OSP)

In the control group, all investigated parts of the small intestine contained SOM-IR neurons. Their numbers were relatively higher in the outer submucosal plexuses (4.50 ± 0.98% in the duodenum, 3.52 ± 0.78% in the jejunum, and 4.23 ± 1.21% in the ileum) (Figure 4B and Figure 5G–I). In the inner submucosal plexuses, the number of SOM containing neurons was visibly lower (0.60 ± 0.06% in the duodenum; 2.36 ± 0.89% in the jejunum, and 2.50 ± 0.44% in the ileum) (Figure 4C and Figure 5M–O). In the experimental group, changes were generally expressed by an increase in the number of SOM-IR neurons in all parts of the intestine as well as in both plexuses studied (Figure 4B,C and Figure 5J–L,P–S). The most significant changes in the number of SOM-IR perikarya were observed in the OSP, both in the duodenum (to 10.20 ± 1.58%) and the jejunum (to 9.25 ± 2.14%) In turn, in the ileum, the changes were less pronounced (an increase to 11.23 ± 1.93%) (Figure 4B). A similar situation occurred in the case of ISP where an increase of SOM-IR neurons was also visible (to 1.25 ± 0.45% in the duodenum, 6.78 ± 1.25% in the jejunum, and 9.60 ± 2.02% in the ileum) (Figure 4C).

### 3.4. Descending Colon

#### 3.4.1. Myenteric Plexus (MP)

In the animals without diabetes, the number of SOM-IR amounted to 2.57 ± 0.16% of all Hu C/D-IR neurons and did not show statistically significant changes in the experimental group (Figure 6 and Figure 7A,D).

#### 3.4.2. Inner and Outer Submucosal Plexuses (ISP and OSP)

In the control group, the number of cell bodies containing SOM was estimated at 3.90 ± 0.52% in the ISP and 2.57 ± 0.16% in the OSP. During diabetes, the total number of SOM-IR perikarya decreased (to 0.27 ± 0.08% in the ISP and 1.54 ± 0.25% in the OSP) (Figure 6 and Figure 7B,C,E,F).

## 4. Discussion

The obtained results are focused on the distribution and number of neurons containing SOM located in the gastrointestinal tract. Moreover, the influence of high glucose serum level induced by streptozocin infusion on the population of SOM-IR intramural neurons was investigated. During the research, it was noted that SOM positive neurons were present in all investigated parts of the porcine gut and corpus of the stomach and were not detected in the antrum of the stomach. However, between particular parts of the GI tract, as well as among particular plexuses, visible differences were noted. In control animals in all of the GI tract regions studied, the total number of SOM-IR neurons generally did not exceed 5% of Hu C/D immunoreactive neurons. In the available literature, there have been several papers reporting the distribution of SOM in the GI tract, mainly in the small intestine [30,31].

In general, the expression of somatostatin in the gastrointestinal tract has been studied in various animal species and in humans [31,32,33,34,35,36]. One of the first animal species in which the distribution of SOM in the digestive tract was described was the guinea pig [14]. In this rodent, the largest number of SOM-IR neurons occur in the myofascial ganglia of the small intestine, while the submucous ganglia contained more than three times less. In addition, for this species, other authors describe other values, as high as 30% in the OSP plexus [14,34]. In the case of studies performed on pigs, the number of SOM positive neurons in individual ganglia is different from that obtained in rodents [20,21,36]. Generally, regardless of the gastrointestinal tract segment, the largest amount of SOM was recorded in the ganglion of the OSP, intermediate in the ganglion of the muscle, and decreased in the ISP. In this study, with the exception of the descending colon, where a large number of SOM neurons were recorded in the ISP in the remaining sections of GI, the number of SOM in individual plexuses corresponded to the above scheme. A similar pattern of quantitative distribution of SOM neurons in individual ganglia of the descending colon of pig results in studies performed by Gonkowski and Calka [10]. However, Pidsudko et al. found that for pig small intestines, the results differ from those presented above [20]. The question arises as to why there are significant differences both between and within species in the case of SOM. In the case of neuropeptides, including SOM, there are several reasons for this observation, for example, tissue fixation time could be of significant importance. According to the biological factors, the expression of a given peptide is influenced by the intestinal microflora, which is different in herbivorous and carnivorous animals. The composition of food mix, stress, and environmental conditions can also have a decisive impact.

Studies conducted during recent years have demonstrated the participation of enteric neurons in many pathological conditions occurring in the GI tract [37,38,39]. One of the most common ways of neuronal adaptation to pathological stimuli is the increased or decreased expression of active substances in neuronal cell bodies [36,38,39]. The results of the current study revealed the noteworthy response of SOM-IR enteric neurons to chemically induced diabetes. Nevertheless, the level of response expressed by the number of neurons synthesizing SOM depended on the particular plexus and region of the GI investigation. In the stomach, the response was restricted to the OSP in the corpus. Regarding the small intestine, there was a very clear increase in the number of SOM-IR cell bodies, especially in both submucosal plexuses. The reverse situation was observed in the descending colon, where diabetes decreased the amount of SOM, notably in the OSP and ISP. The amount of available information concerning the changes in SOM immunoreactivity in the GI tract in the course of diabetes is scarce [36]. In hyperglycemic diabetic rats, a reduction in the SOM-containing neurons in the small intestine was noted [23]. Unfortunately, there are no data on the stomach and the descending colon. Furthermore, the above-mentioned studies were based on rat models of diabetes. Moreover, the above authors used BB rats, and although this animal model is appropriate for the study of pathogenesis of type I diabetes, it is not appropriate for investigating enteric neuropathy followed by a high glucose level [37]. Obviously, altered SOM enteric neurons activity also accompanied several other well-understood clinical conditions. For example, a decrease in SOM immunoreactivity has been noted during colonic cancer, infection *Bacteroides fragilis*, and proliferative enteropathy [20,21]. In turn, an increase in SOM positive neurons has been observed under chemically induced colon inflammation and through proliferative enteropathy in the porcine ileum. The current results confirm previous observations regarding the plasticity of SOM positive enteric neurons, although this variability strongly depended on the alimentary tract region and differed between particular plexuses.

Despite much effort being focused on functional aspects of SOM activity in the GI tract, many functions have not yet been fully clarified. One of the best-known actions of SOM is its inhibitory effect [39]. This refers to two major gastrointestinal functions, i.e., inhibition of motor activity and secretory processes [40,41]. The influence of SOM on contraction activity is mediated via the inhibition of acetylcholine release [42]. The physiological consequences of SOM action are a delay in gastric emptying and food retention. In many patients with chronic diabetes, gastrointestinal disturbances have been described [43]. One of the dominant disorders is post-prandial fullness [43]. The current results suggest that SOM, on the one hand, through an increase in the number of neurons synthesizing SOM, especially in the stomach corpus and small bowel, is at least partially responsible for these symptoms. On the other hand, the decreased number of SOM neurons in the descending colon observed in the current study may be responsible for unpleasant symptoms, such as faecal incontinence or diarrhea. Furthermore, anti-inflammatory and anti-nociception properties of SOM were also described and appear to be important in the pathogenesis of gastric complications of diabetes. Diabetes is accompanied by high oxidative stress conditions that lead to immune cell activation and the augmentation of pro-inflammatory factor synthesis [43,44]. Thus, inflammation can be the cause of enteric nerve damage leading to gastrointestinal motility disorders. Furthermore, abdominal pain is another symptom that can occur in diabetic patients with enteric neuropathy [45]. SOM can be involved in both of the above-mentioned processes. It is probably released outside of the cell body and decreases pro-inflammatory cytokine expression and release, as well as lymphocyte proliferation. This action of SOM was confirmed by its analogues administered exogenously, which are known to exhibit systemic anti-inflammatory and anti-nociceptive effects [45]. According to the role of SOM described above, it can be supposed that this peptide plays a pivotal role in the swine model of hyperglycemia. An increase in SOM-IR neurons in the submucosal plexus in the small intestine and in the stomach corpus, as well as in the myenteric plexus of the ileum, are a response to the toxic effect of glucose and probably reduces the effect of abdominal pain and inflammatory conditions. However, excessive amounts of SOM-IR neurons can lead to contractility disturbances [36,46].

## 5. Conclusions

In summary, these results could indicate the important role of SOM in GI tract pathology caused by hyperglycemia. The exact function of SOM in this process is not fully known, but the current results provide immunohistochemical evidence that SOM is another neuropeptide that undergoes changes in the course of hyperglycemia. Although analogues of somatostatin are currently being tested as virtual drug treatments for GI tract complications, further work is needed to elucidate the detailed action of somatostatin.

## Figures and Tables

**Figure 1 animals-10-00142-f001:**
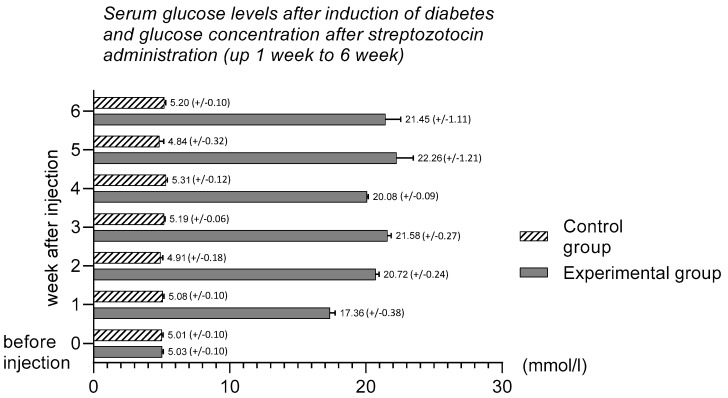
Serum glucose concentration before induction of diabetes and glucose accumulation after streptozotocin injection (up 1 week to 6 weeks).

**Figure 2 animals-10-00142-f002:**
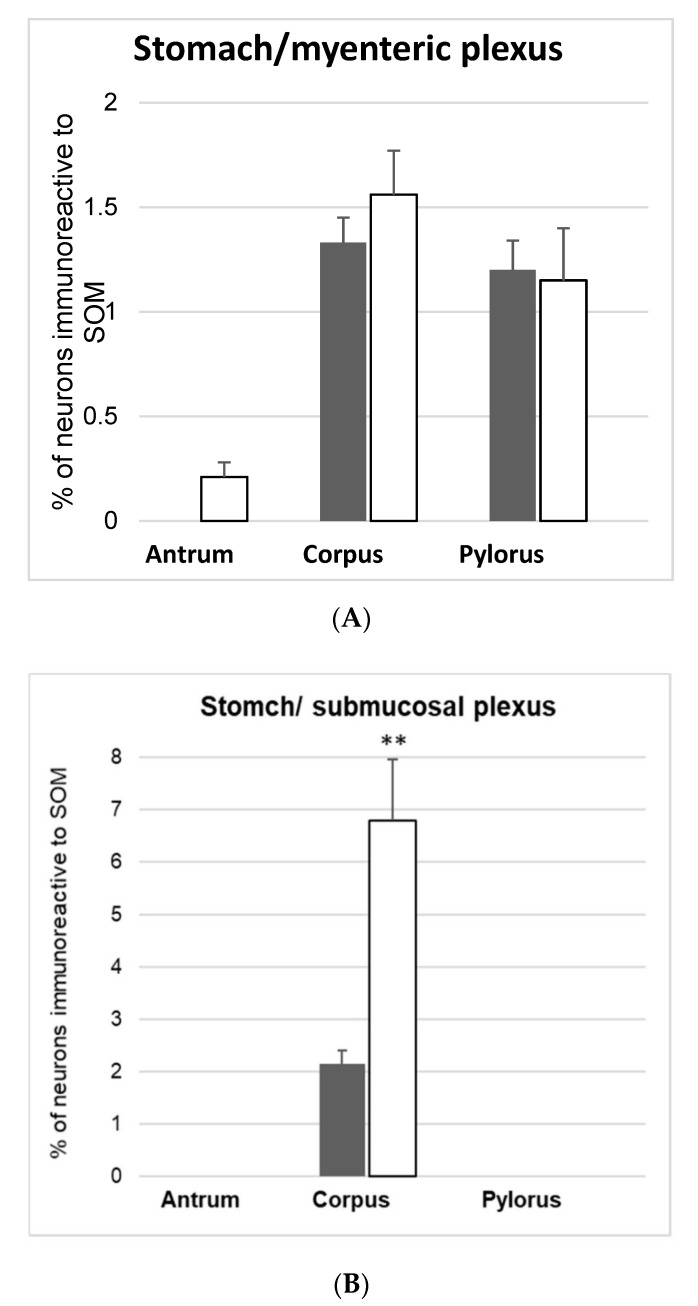
Mean (±SEM) of SOM, like immunoreactive (SOM-IL), neurons in the myenteric plexus (**A**) and submucosal plexus (**B**) of stomach in the control (white bars) and streptozotocine-induced diabetes groups (grey bars). Data are presented as mean ± SEM and statistically significant data (* *p* < 0.05, ** *p* < 0.01, and *** *p* < 0.001).

**Figure 3 animals-10-00142-f003:**
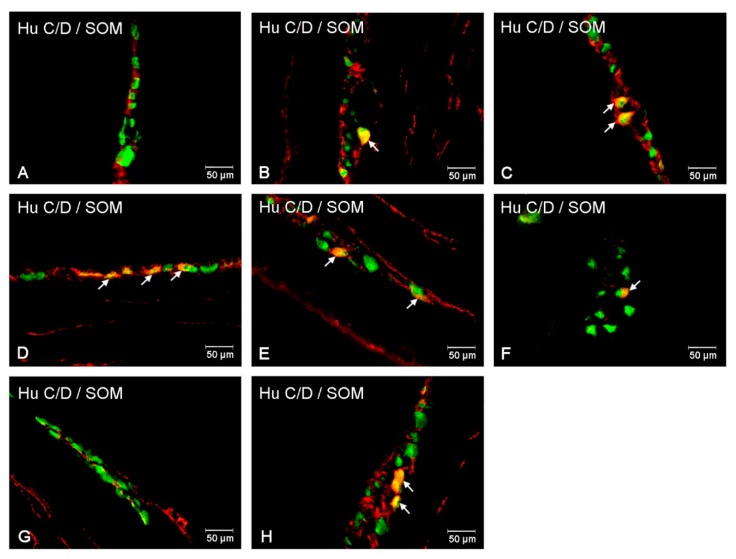
Intramural plexuses in the porcine stomach (antrum, stomach, and pylorus) immunoreactive to SOM. Myenteric plexus of the antrum (**A**), corpus (**B**), and pylorus (**C**) under physiological condition (**A**–**C**), and after streptozotocin administration (**D**–**F**). Submucosal plexus of the corpus (**G**), under physiological state and after streptozotocin injection (**H**). All photographs have been created by digital superimposition of two color channels (green for Hu C/D used here as a pan-neuronal marker, and red for SOM). Neurons showing co-localization of Hu C/D and SOM are indicated with arrows.

**Figure 4 animals-10-00142-f004:**
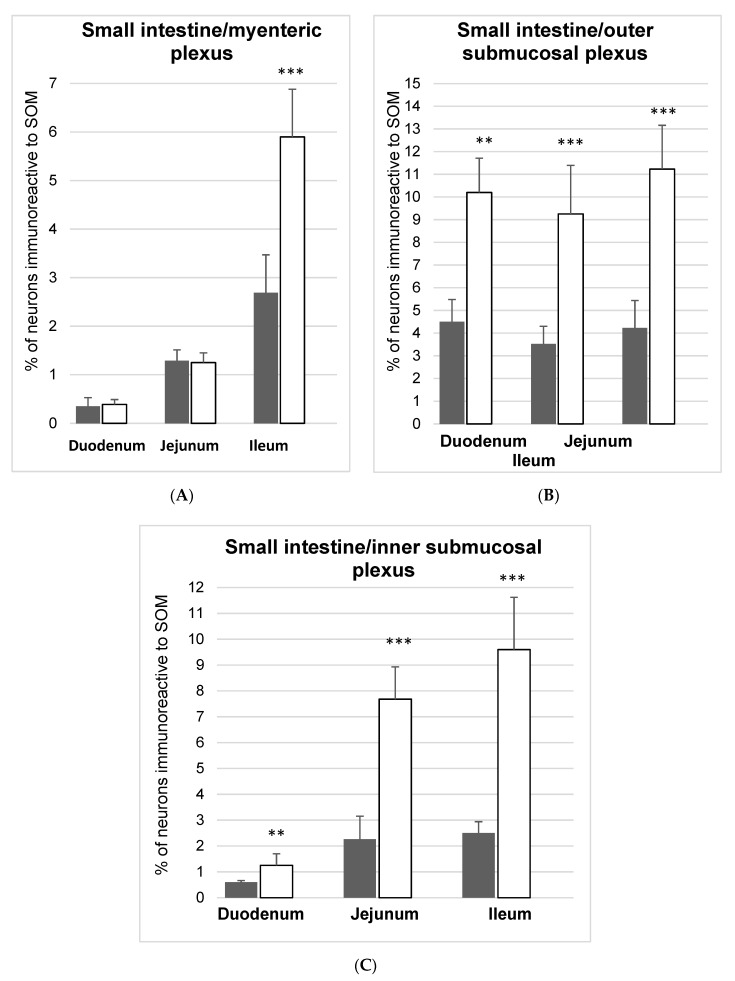
Mean (±SEM) of SOM, like immunoreactive (SOM-IL), neurons in the myenteric plexus (**A**), outer submucosal plexus (**B**) and inner submucosal plexus (**C**) of the small intestine in the control (grey bars) and streptozotocine-induced diabetes groups (white bars). Data are presented as mean ± SEM and statistically significant data (* *p* < 0.05, ** *p* < 0.01, and *** *p* < 0.001).

**Figure 5 animals-10-00142-f005:**
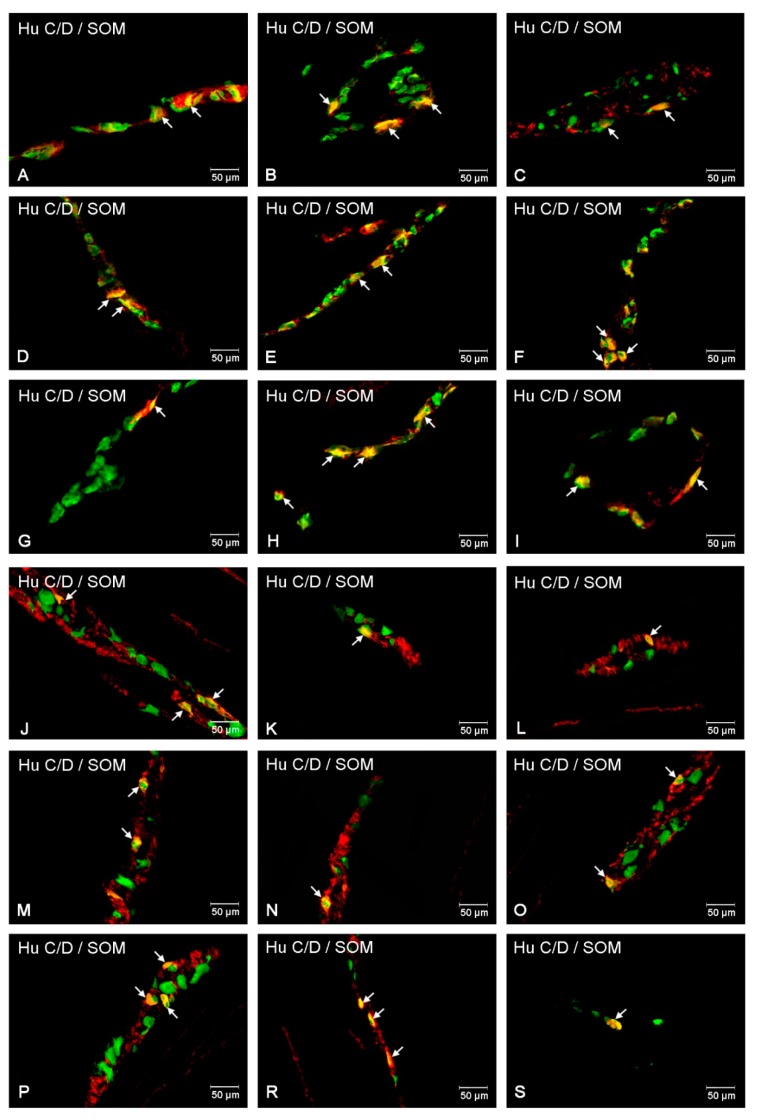
Intramural plexuses in the small intestine immunoreactive to SOM. Myenteric plexus of duodenum (**A**,**D**), jejunum (**B**,**E**), and ileum (**C**,**F**) under physiological condition (**A**–**C**), and hyperglycemic animals (**D**–**F**). Outer submucosal plexus of the duodenum (**G**–**J**), jejunum (**H**–**K**), and ileum (**I**–**L**) under physiological state (**G**–**I**) and after streptozotocin injection (**J**–**L**). Inner submucosal plexus of the duodenum (**M**), jejunum (**N**) and ileum (**O**) under physiological condition (**M**–**O**) and after streptozotocin administration (**P**–**S**). All photographs were created by digital superimposition of two color channels (green for Hu C/D used here as a pan-neuronal marker, and red for SOM). Neurons showing co-localization of Hu C/D and SOM are indicated with arrows.

**Figure 6 animals-10-00142-f006:**
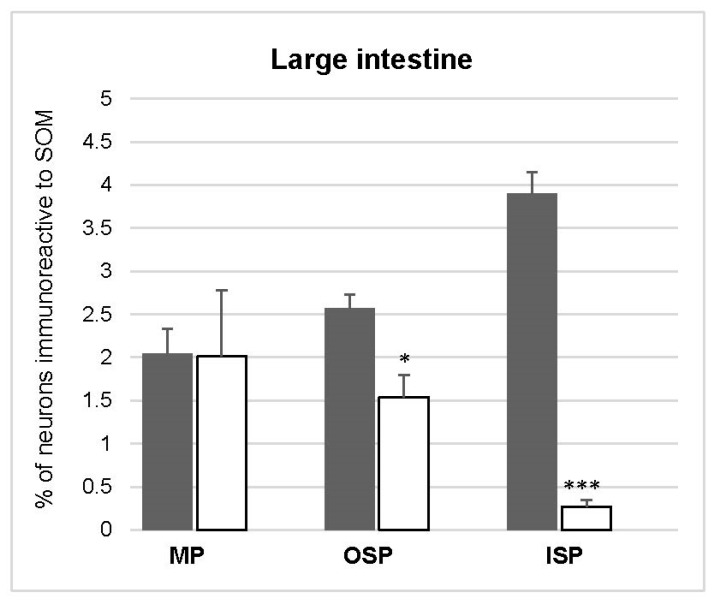
Mean (±SEM) of SOM, like immunoreactive (SOM-IL), neurons in the myenteric plexus, outer submucosal plexus, and inner submucosal plexus of large intestine in the control (white bars) and streptozotocine-induced diabetes groups (grey bars). Data are presented as mean ± SEM and statistically significant data (* *p* < 0.05, ** *p* < 0.01, and *** *p* < 0.001).

**Figure 7 animals-10-00142-f007:**
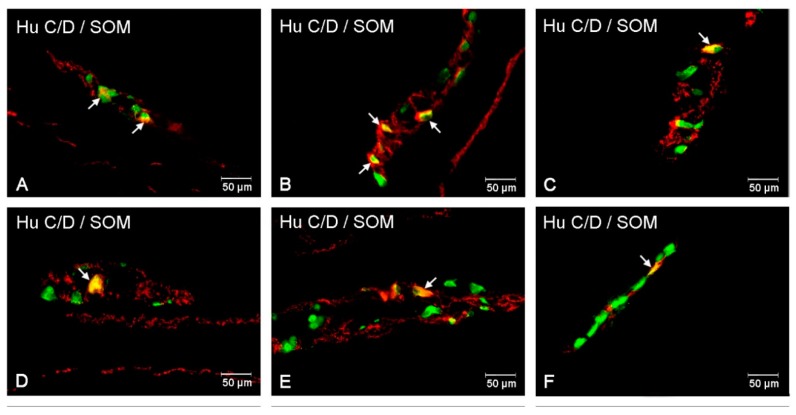
Intramural plexuses in the large intestine part expressing SOM. Myenteric plexus of descending colon under physiological condition (**A**) and hyperglycemic animals (**D**). Outer submucosal plexus of the descending colon in the control group (**B**) and in experimental animals (**E**). Inner submucosal plexus of the descending colon in control animals (**C**) and in diabetic animals (**F**). All photographs have been created by digital superimposition of two color channels (green for Hu C/D used here as a pan-neuronal marker, and red for SOM). Neurons showing co-localization of Hu C/D and SOM are indicated with arrows.

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
