# Peer review of "The Influence of a Hyperglycemic Condition on the Population of Somatostatin Enteric Neurons in the Porcine Gastrointestinal Tract"

_animals, 2020, doi:10.3390/ani10010142_

Round 1

Reviewer 1 Report

It seems that the authors sufficiently adressed for this reviewer's comments. This reviewer has only two minor additional comments.

Figure 2: Based on the line 195, SOM-IR somata increased to 6.78%, however, the vertical line of Fig. 2B indicates up to 2%. Numerical numbers of the vertical line may not  be correct.  Figure 2, figure 4 and figure 6: By figure legents, white bars indicate control groups and grey bars diabetes groups. However, for example, Fig.4C indicates that white bars represent the increase of SOM-IR. Explanation of white and grey bars may not be correct.

Author Response

Reviewer 1

Figure 2: Based on the line 195, SOM-IR somata increased to 6.78%, however, the vertical line of Fig. 2B indicates up to 2%. Numerical numbers of the vertical line may not  be correct.  Figure 2, figure 4 and figure 6: By figure legents, white bars indicate control groups and grey bars diabetes groups. However, for example, Fig.4C indicates that white bars represent the increase of SOM-IR. Explanation of white and grey bars may not be correct.

Authors comments

 Thank you for your comment. Authors will add this suggestion. 

Reviewer 2 Report

Although most concerns have been adequately addressed, a couple of concerns remain.

Major comments,

With respect to Figure 1, previously published data should not be submitted without permission. It is a duplicate submission.

Minor comments,

In Abstract,

Line 40, abbreviation “(ENS)” should be deleted.

In Introduction,

Line 104, “somatostatin” should be abbreviated as “SOM”.

In Material and methods,

Line 136, abbreviation “PB” should be added; e.g. “phosphate buffer (PB; pH 7.4)”. Line 151 and 154, “x” should be changed to “×”. Line 164, “as a mean ± standard error SEM of mean” should be changed to “as a mean ± SEM”.

In Results,

Figures 2A and 2B are the same. Line 228, the statement “the changes were less pronounced” is not correct. In the ileum, the number of SOM-IR neurons is increased after treatments of streptozotocin (control groups: 4.23 ± 1.21%; streptozotocin-induced diabetes groups: 11.23 ± 1.93%). In Figure 4A and B, asterisks should be added to the graph. Line 248, “7 Fig A, D” should be changed to “Fig 7A, D”. Line 251, the value in the text (4.23 ± 1.21% in the OSP) is different from that of the graph of Figure 6 (2.57 ± 0.16%).

Author Response

Major comments,

With respect to Figure 1, previously published data should not be submitted without permission. It is a duplicate submission.

Authors comments

Presented results are a part of different studies which were performed on diabetic pigs. The aim of these study was examined and described influences if hyperglicemia on chemical coding on enteric neurons in different parts of gastrointestinal tract.  Due to high number of investigated substances as well as big amount of digestive tract sections results are presented in several publications. Due to ethical, legal reasons as well as ethical committee recommendation the number of animals using in experimental study according to Poland a European law should be limited to a minimum (rule 3R). In the case of our studies we received permeation on 10 animals (n=5 in control and n=5 in experimental). In previous publications we always provide information concerning blood level in animals. But this information is given only as a confirmation that all animals in experimental group developed diabetes and this information is not goal of this study and these are not results.  Moreover, information about glucose level presented in previous study was also included in results chapter and we quote appropriate articles.

Minor comments,

In Abstract,

Line 40, abbreviation “(ENS)” should be deleted.

Authors comments

It will be corrected

In Introduction,

Line 104, “somatostatin” should be abbreviated as “SOM”.

Authors comments

It will be corrected

In Material and methods,

Line 136, abbreviation “PB” should be added; e.g. “phosphate buffer (PB; pH 7.4)”. Line 151 and 154, “x” should be changed to “×”. Line 164, “as a mean ± standard error SEM of mean” should be changed to “as a mean ± SEM”.

Authors comments

It will be corrected

In Results,

Figures 2A and 2B are the same. Line 228, the statement “the changes were less pronounced” is not correct. In the ileum, the number of SOM-IR neurons is increased after treatments of streptozotocin (control groups: 4.23 ± 1.21%; streptozotocin-induced diabetes groups: 11.23 ± 1.93%). In Figure 4A and B, asterisks should be added to the graph. Line 248, “7 Fig A, D” should be changed to “Fig 7A, D”. Line 251, the value in the text (4.23 ± 1.21% in the OSP) is different from that of the graph of Figure 6 (2.57 ± 0.16%).

Authors comments

It will be corrected

Reviewer 3 Report

I appreciate that the authors have addressed many of the concerns highlighted in my first review, in particular addition of references and description of SOM distribution in the pig GIT has added much value to the text, a few small changes are still required.

I still believe that the conclusion/summary is an overstatement of the results.

Line 348; The exact function of SOM in this process is not fully known, but the current results provide immunohistochemical evidence that SOM is another neuropeptide that can affect motor and resorption processes in the course of hyperglycaemia.

This is not correct, your study shows that SOM expression is altered in pigs in a hyperglycaemic state, it does not show the role of SOM in motor or resorption processes. This sentence needs to be removed or worded.

Author Response

Reviewer 3

I appreciate that the authors have addressed many of the concerns highlighted in my first review, in particular addition of references and description of SOM distribution in the pig GIT has added much value to the text, a few small changes are still required.

I still believe that the conclusion/summary is an overstatement of the results.

Authors comments

In these section authors provide short information about SOM expressed in enteric neurons in the course of diabetes and do not repeat information presented in results moreover sentence concerning motor activity will be removed.

Line 348; The exact function of SOM in this process is not fully known, but the current results provide immunohistochemical evidence that SOM is another neuropeptide that can affect motor and resorption processes in the course of hyperglycaemia.

This is not correct, your study shows that SOM expression is altered in pigs in a hyperglycaemic state, it does not show the role of SOM in motor or resorption processes. This sentence needs to be removed or worded.

Authors comments

It will be corrected
